# Physiotherapist beliefs and perspectives on virtual reality supported rehabilitation for the management of musculoskeletal shoulder pain: A focus group study

Niamh Brady [1,2]*, Beate Dejaco[3,4], Jeremy Lewis[5,6], Karen McCreesh[7,8], Joseph G. McVeigh[1]

1 Discipline of Physiotherapy, University College Cork, Cork, Ireland, 2 Evolve Health, Cork, Ireland, 3 Delta Sports Medical Centre Papendal, Arnhem, The Netherlands, 4 HAN University of Applied Sciences, Radboudumc, Nijmegen, The Netherlands, 5 School of Health and Social Work, University of Hertfordshire, Hatfield, Hertfordshire, United Kingdom, 6 Therapy Department, Central London Community Healthcare National Health Service Trust, London, United Kingdom, 7 School of Allied Health, University of Limerick, Limerick, Ireland, 8 Ageing Research Centre, Health Research Institute, University of Limerick, Limerick, Ireland

* niamh.brady@ucc.ie

**Data Availability Statement:** All relevant data are within the paper and its Supporting Information files.

## Abstract

### Background

Virtual Reality (VR) is an immersive computer-generated environment that provides a multi-sensory experience for the user. Modern technology allows the user to explore and interact with the virtual environment, offering opportunities for rehabilitation. The use of immersive VR in the management of musculoskeletal shoulder pain is relatively new and research is required to demonstrate its feasibility and effectiveness in this field.

### Aim

The aims of this study were, firstly, to explore physiotherapists' beliefs and perspectives of immersive VR as a platform for rehabilitation in patients with musculoskeletal shoulder pain, secondly, to identify potential barriers and facilitators to using VR in a musculoskeletal set-ting and thirdly, to gain insight from clinicians that would inform the development of a VR intervention for the rehabilitation of musculoskeletal shoulder pain.

### Methods

This study used qualitative descriptive design methodology. A series of three focus group interviews were carried out, via Microsoft Teams. Physiotherapists received an Oculus Quest™ headset to use at home prior to the focus group interviews. A six-phase process of reflexive thematic analysis was carried out to identify themes within the data. Atlas Ti Quali-tative Data Analysis software was used to facilitate thematic analysis.

**Funding:** The authors received no specific funding for this work.

**Competing interests:** The authors have declared that no competing interests exist.

## Results

Five themes were identified within the data. They reflected physiotherapists beliefs that VR provides novel opportunities for shoulder rehabilitation and may offer new avenues for managing movement-related fear and improving concordance with rehabilitation. However, barriers related to safety and practical considerations associated with using VR were also identified in the final themes.

## Conclusion

These findings provide valuable insight into clinician acceptability of immersive VR as a platform for rehabilitation and the need for further research to answer the questions posed by physiotherapists in the current study. This research will contribute to human-centered design of VR-supported interventions for managing musculoskeletal shoulder pain.

## Introduction

Shoulder pain is common and is often associated with substantial disability [1]. The lifetime prevalence of shoulder pain is 70% [2] and between 25% and 70% of those who suffer with shoulder pain will continue to experience symptoms 1-year post onset [3–5]. Understanding the source of shoulder pain is complex and in many cases, observed pathoanatomical changes do not adequately explain symptoms [6]. A biopsychosocial approach in the management of musculoskeletal shoulder conditions is paramount and includes prioritizing shared decision making for those people seeking care [7]. For most individuals with shoulder pain a non-surgical management approach is appropriate, and a graduated exercise program should form a part of overall management [8–14]. Although we do not know what type or intensity of exercise is best, several systematic reviews support specific over generic exercise therapy [15–17] and progressive, resisted exercise over non-progressive, non-resisted exercise [18, 19]. No one exercise program has demonstrated definitive superiority in the management of shoulder pain [20, 21]. Poor concordance to exercise-based interventions may be a barrier to successful rehabilitation outcomes [22, 23]. Concordance is influenced by co-morbidities, psychological factors, pain, and boredom while carrying out prescribed exercise [22, 24, 25].

Virtual reality (VR)-based interventions have been shown to be effective in managing acute and persistent pain in various settings and clinical populations, including musculoskeletal pain [26–30]. VR technology offers an opportunity to support exercise prescription in the rehabilitation of musculoskeletal shoulder pain. The mechanisms by which VR facilitates pain management are unknown but may include distraction [31], manipulation of somatosensory input [32, 33], and graded exposure [34]. Hoffman et al [35, 36], utilized functional magnetic resonance imaging (MRI) to investigate the effect of VR on experimental pain and showed that there was a reduction in activity in five areas of the brain that are associated with pain. Thomas et al, [34] demonstrated that a VR dodgeball game, designed to encourage individuals with low back pain and kinesiophobia to increase lumbar spine flexion through gameplay, was well tolerated and led to an increase in movement during and between sessions. Similarly, Hennessy et al, [37] demonstrated the safety and acceptability of a VR application designed to facilitate graded exposure to everyday tasks for individuals with low back pain (LBP).

Evidence suggests that VR may be an attractive platform for rehabilitation of neurological and musculoskeletal conditions [30, 38–40]. This may be because of the potential of VR to enhance motivation and adherence to exercise programmes [40] and to educate people about

various health conditions [41, 42]. Using VR to educate people may be a useful tool since management of musculoskeletal conditions requires an active approach to rehabilitation and lifestyle change [7]. Research has shown that in healthy participants, VR has the potential to increase metabolic requirements associated with physical activity, including oxygen consumption and heart rate [43, 44]. Since physical activity plays such an important role in musculoskeletal health, VR may have the potential to improve outcomes from therapeutic exercise programmes and physical activity interventions.

There are different types of VR interventions, ranging from non-immersive to fully immersive technology. Non-immersive VR involves the user observing a computer-generated image of a virtual environment on a screen, but the user is not immersed completely in the virtual world. Immersive VR involves the use of a head mounted display (HMD) unit that offers a multi-sensory experience for the user. Additional equipment such as hand-held devices, gloves, and vibrotactile platforms may provide sensory feedback to the user, enabling them to explore and interact with the virtual environment as an avatar. Hoffman et al, [45] suggest that more immersive high-tech VR providing a greater illusion and taking more conscious attention may lead to a greater reduction in pain than low-tech non-immersive VR. As such, they investigated the dose-response relationship between high-tech VR and low-tech VR. Participants in the high-tech VR group reported significantly greater "fun" and a stronger sense of illusion. Post-VR pain thresholds correlated with the stronger sense of illusion, suggesting a greater sense of embodiment (high-tech VR) is associated with a reduction in the experience of pain more than a reduced sense of embodiment (low-tech VR).

Research investigating the role of immersive VR interventions for managing musculoskeletal conditions is growing and is associated with promising results [30]. In a randomized controlled trial (RCT) by Bahat et al [46] comparing VR supported rehabilitation to standard rehabilitation for neck pain, only those who used VR for rehabilitation showed clinically important improvements in Neck Disability Index (NDI) scores immediately post-intervention and at three months follow up [46–48]. In an RCT including individuals with fibromyalgia, when VR was added to standard rehabilitation (which comprised of aerobic exercise and Pilates-based exercise), significantly greater improvements in pain and kinesiophobia were found [49]. Another RCT has demonstrated that a home-based VR intervention, which involved behavioural skills training and pain education, was effective for managing chronic LBP [50]. In this study, the immersive VR intervention successfully reduced pain intensity and pain interference with activity, mood and stress compared to a non-immersive VR sham intervention. For the VR group, large pre-post effect sizes met moderate to substantial clinical importance for reduced pain intensity and pain-related interference with activity, mood, and stress. A small number of studies have investigated the effectiveness of non-immersive VR for managing musculoskeletal shoulder pain, specifically subacromial pain syndrome [51] and frozen shoulder [52]. Pekyavas and Ergun [51] randomized people with subacromial pain to treatment with either VR or standard rehabilitation and found that both groups showed statistically significant improvements in pain intensity. When pre-post treatment Shoulder Pain and Disability Index (SPADI) scores were compared, the VR group showed a statistically and clinically important reduction compared to a non-significant reduction in the exercise group. When measurements were taken at one-month following the intervention, both groups displayed clinically and statistically significant improvements in disability.

Immersive VR technology is evolving rapidly, which has implications for the quality of the user's experience but also for safety and feasibility. Modern technology is lightweight and comfortable to wear, has improved motion tracking and minimal visual latency, which reduces the risk of experiencing motion sickness. Therefore, research using contemporary VR systems is needed to explore the utility of VR in musculoskeletal conditions, such as shoulder pain.

Birckhead et al [53] suggest that VR intervention development should begin with direct input from both provider and patient end-users to optimize human-centered design. In this way, it may be possible to determine the acceptability of the intervention and identify potential barriers and facilitators which can then be addressed early. By addressing acceptability at an early stage, the intervention can be adapted and therefore participation may increase [54].

## Study aims

The aims of this study were, firstly, to explore physiotherapists' beliefs and perspectives of immersive VR as a platform for rehabilitation in patients with musculoskeletal shoulder pain, secondly, to identify potential barriers and facilitators to using VR in a musculoskeletal setting and thirdly, to gain insight from clinicians that would inform the development of a VR intervention for the rehabilitation of musculoskeletal shoulder pain.

## Methodology

This study used qualitative descriptive design to explore physiotherapists' beliefs and perspectives about using immersive VR for the rehabilitation of individuals with shoulder pain. Qualitative descriptive methodology was deemed appropriate for this research question as it aims to provide a description and detailed summary of the phenomenon of interest using participants' own language [55]. Low-inference interpretation during data analysis means that the researcher stays close to the data [56] and this process provides preliminary insight into novel clinical interventions, like VR. Qualitative description is a flexible methodology for data collection and analysis [56] and results in insight and information that is both broad and rich. Qualitative description has been described as an "excellent choice for the healthcare environments designer, practitioner or health sciences researcher because it provides rich descriptive content from the subject's perspective"[57]. This study is reported according to the Consolidated Criteria for Reporting Qualitative Research (COREQ) checklist [58].

## Participants and recruitment

Participants were clinical physiotherapists who work with musculoskeletal shoulder pain on a regular basis. Purposive sampling was used to recruit physiotherapists with varying levels of clinical experience and from a range of clinical work environments. Physiotherapists with variation in socio-demographic working environments and across public and private sectors were recruited as well as physiotherapists with and without previous experience of using immersive VR in any capacity (clinical or entertainment).

Participants were recruited from acute hospitals, community hospitals, primary care centres and private practices in Ireland using various recruitment methods. A study information sheet was sent to individual physiotherapy departments and private practices in Ireland. Professional bodies, including the Irish Society of Chartered Physiotherapists (ISCP), and relevant clinical interest groups including the Irish Shoulder and Elbow Rehabilitation Society (ISERS) were contacted and invited to distribute study information to members via email. The study was also promoted on social media platforms; Instagram™ and Twitter™. The study information sheet outlined the study background, aims and design. Physiotherapists who were interested in participating in the study, contacted the Primary Investigator (PI, NB) by telephone or email. The PI screened potential participants by telephone to assess appropriateness for inclusion in the study. Physiotherapists who met the inclusion criteria (having a minimum one-year clinical experience with a minimum six months' experience working with musculoskeletal pain conditions; shoulder pain presentations accounting for minimum 10% of overall caseload; English-speaking) were sent a confirmation email with consent form attached.

## Data collection

A series of three focus group interviews were carried out, via Microsoft Teams, with physiotherapists based in the Republic of Ireland. Focus groups were chosen as the method for data collection as they allow for the emergence of important themes that may be overlooked in individual interviews with a more structured question schedule [59]. Focus groups facilitate the exploration of both shared and differing views between participants and have been identified as an appropriate method for informing intervention development [60]. After consenting to the study and prior to the focus group interviews, physiotherapists received an Oculus Quest™ headset to use at home for one week (one physiotherapist had an Oculus headset at home for personal use and therefore did not require and additional headset). Participants were asked to use the headset at least three times for a maximum 20-minute duration in each session. Oculus Quest health and safety warnings advise to use a headset for maximum 30-minute duration. However, since all participants were new to the technology and to minimize the risk of motion sickness and fatigue, it was decided that a 20-minute duration limit would be appropriate for this study. All physiotherapists were sent an informative video to watch at home, demonstrating how to safely use VR. They were instructed to discontinue use if they experienced adverse effects and to inform the research team.

Participants were encouraged to explore Oculus Quest's native First Steps tutorial (https://www.oculus.com/experiences/quest/3675568169182204?ranking_trace=0_3675568169182204_QUESTSEARCH_28b8d286-6206-4581-8c2c-c876d55e6943) which shows the user how to safely set up the VR environment and how to use the hand-held controllers for manipulating objects in the virtual world. Participants also had the opportunity to try various demonstration activities which are available on Oculus Quest including a popular VR rhythm game called Beat Saber ™ (https://www.oculus.com/experiences/quest/1758986534231171?ranking_trace=0_1758986534231171_QUESTSEARCH_d0b1e295-5201-4bde-a1cb-8d2440c61b2f), and a sports game called Sports Scramble ™ (https://www.oculus.com/experiences/quest/2131072803612066?ranking_trace=0_2131072803612066_QUESTSEARCH_009ba037-c88a-4b42-9363-441abbdac775). Therefore, participants had the opportunity to experience immersive and interactive VR and learned to use their virtual hands to move objects, play virtual sports, punch an opponent in a boxing match, dance and explore a range of virtual worlds.

Within two months of using the headset, participants were invited to participate in an online focus group interview, conducted using MS Teams. Each interview lasted approximately 90 minutes. These interviews took place between March 2021 and March 2022. Focus group interviews were video recorded with the consent of all participants. Two members of the research team facilitated the interviews. The PI (NB) used a semi-structured question schedule (S1 File) to guide the interview while an additional member of the research team (BD) ensured that the interview was being recorded, took notes and on occasion probed participants with follow-up questions. The question schedule was flexible and was adapted and expanded over the course of the data collection process, to allow for the exploration of new ideas and themes suggested by participants [61]. The question schedule was designed to explore physiotherapists perspectives on whether VR is a feasible intervention for use with individuals who have shoulder pain, as well as anticipated barriers and facilitators to the use of VR technology in clinical practice. Interview questions were piloted with two physiotherapists not included in the study to check for clarity and comprehensibility. Interviews were informal and participants were encouraged to converse with one another and to contribute as much as they felt comfortable, on a first name basis.

## Data analysis

Data collection and analysis occurred concurrently to allow early identification of codes and themes and to determine data saturation. Data saturation was considered when in the third focus group interview, minimal additional codes emerged and those that did contributed little to the overall discussion. The first two interviews were transcribed manually. The final interview was transcribed using Microsoft Teams' native transcription function. Interview transcripts were analyzed by two members of the research team (NB and BD). A six-phase process of reflexive thematic analysis (see Table 1) was carried out to identify "patterns or themes within data" [62–65]. Atlas Ti Qualitative Data Analysis software was used to facilitate thematic analysis. A framework described by Nowell et al [66] was used to ensure trustworthiness in each phase of data analysis. The framework included several strategies such as: prolonged engagement with the data; data triangulation; peer debriefing; reflexive journaling; researcher triangulation, member checking and a clear description of the audit trail. This framework was used to achieve trustworthiness as described by Lincoln and Guba [67] as credibility, transferability, dependability and confirmability. After each phase of data analysis NB debriefed the remaining members of the research team on progress and all contributed to the analysis and sorting of codes and generation of initial themes. The full research team were involved in the review and refinement of final themes and in the preparation of the final report. When data analysis was complete, final themes and a summary of the results were shared with participants via email to check that they did indeed reflect the views of participants.

## Reflexive practice

Research team members (NB and BD) participated in a reflexive practice, prior to data collection and following each focus group interview. A reflection diary was used to document individual researchers' professional backgrounds and relationship to the research topic. This included personal biases and initial thoughts around codes and themes. Niamh Brady (BSc, MPhty) is a musculoskeletal physiotherapist working part-time in clinical practice at (12 years' experience) and part-time as lecturer in anatomy in University College Cork, Ireland. She is currently undertaking a PhD. This is her first piece of research using qualitative methodology, after having completed three days of training in qualitative data enquiry at University College Cork. Beate Dejaco (FT, SFT, MT MSc) is a senior musculoskeletal physiotherapist, working part-time in clinical practice (23 years' clinical experience) and part-time as lecturer at HAN

**Table 1. Thematic analysis process, adapted from Braun and Clarke ([63]).**

| Phase | Tasks |
|-------|-------|
| Phase 1 | Familiarisation with the data, including the transcription of video interviews into written text, reading and re-reading of the data, and capturing any initial ideas. |
| Phase 2 | Creation of initial codes, a code being an identifier for some feature of the data that may be of interest and collecting the data that is relevant to those codes. |
| Phase 3 | Analysing and sorting these codes into broader themes and collating the data associated with those codes within these themes. |
| Phase 4 | Reviewing and refining of themes, such as discarding themes without enough data or combining themes. |
| Phase 5 | Further refinement of themes and writing a detailed analysis of each theme. |
| Phase 6 | Producing the final report from the detailed analysis of the themes. |

University of Applied Sciences, Netherlands. She too is currently completing her PhD, and this is her first experience in qualitative research.

Both researchers acknowledged that their pre-existing biases could potentially influence data collection and analysis and were conscious not to let this interfere with a rigorous qualitative research approach. On the other hand, they were aware of how a common professional background with the participants having could facilitate a sense of trust in the interview process and an open discussion around the research topic. Braun and Clarke [65] recognize the researcher's role in knowledge production and that themes are "produced at the intersection of the researcher's theoretical assumptions, their analytic resources and the data themselves".

## Ethical approval

Ethical approval was gained from the UCC Social Research Ethics Committee prior to recruitment of physiotherapists. Informed consent was gained remotely once physiotherapists returned a consent form with electronic or typed signature via email, indicating their wish to participate.

## Results

Participants (n = 16) were organized into three focus groups. Focus group one included five participants, focus group two included six participants and focus group three included five participants. Across three focus group interviews, five themes were identified. Results are presented to reflect the inductive process of qualitative descriptive analysis and the aim of understanding the perspectives of physiotherapists. Therefore, an overall description of the prevalence, strength, and depth of discussion around various themes is provided, rather than a count of the number of times themes were mentioned by participants[68].

Physiotherapists suggested several "opportunities" or areas of "potential" for using VR in managing individuals with shoulder pain.

"I mean within five seconds you can see the potential of this. It just blew my mind" (FG 1, P1).

Physiotherapists imagined how they might use VR technology in clinical practice and specific shoulder conditions that they felt would benefit from VR rehabilitation. Physiotherapists shared their own personal experiences with VR and what VR applications they enjoyed most. They also suggested ways in which specific elements of VR could be utilized to facilitate rehabilitation in a customized shoulder rehabilitation program. Physiotherapists also suggested various barriers that would need to be overcome before they would consider using VR in clinical practice.

### Theme 1: Immersion in VR reduces fear of movement. "When I had the VR headset on, I was rotating, moving, twisting a lot more"

During the focus group discussions, physiotherapists suggested that the immersive quality of VR was a key factor and would encourage a shift of attention away from the painful body part towards a focus on the VR environment. Physiotherapists suggested that being immersed in VR would potentially reduce fear avoidant behavior associated with shoulder pain. One physiotherapist considered distraction as a mechanism at play this instance:

"That nice bit of distraction for people that were maybe a little bit fearful. If you think of your rotator cuff people that don't like lifting their arm overhead. You could see how that little bit of distraction or engagement, in the game, how you might kind of forget about that (moving their arm overhead)." (FG 3, P3)

Two physiotherapists reported experiencing musculoskeletal pain prior to participating in the study and shared valuable insight based on personal experience. In one case, the physiotherapist described having low back pain which was typically aggravated by extension movements. This caused the physiotherapist to be conscious and careful of extension activities. He reported that while using VR, he found himself extending his back in a way that he would not normally be comfortable with. He also reported no resulting pain or flare of symptoms associated with increased movement while using VR.

"When I was playing the space game, I noticed that I was dodging a bullet left and right and I couldn't get over how much extension I was doing. I would have quite a stiff painful back, but I was shooting things and then you know (leans to side) it really was brilliant and I was kinda paying attention to everything else other than the back" (FG 1, P1)

When he was asked if he would have had the same reaction if he was using a non-immersive device such as Nintendo Wii$^{TM}$, he stated:

"Zero. No, it wouldn't have happened. Because the bullet was coming at my head (immersive VR). Whereas on a screen (non-immersive VR) the bullet is.. (holds hands out in front)".

Similarly, a physiotherapist who was recovering from a rib injury shared her experience of using VR. She felt that she was encouraged to move and exercise more than she would have done in her own personal rehabilitation program.

"So I just have a rib injury at the minute, and I know when I'm working out I'm conscious of it. And when I had the VR headset on, I was definitely rotating, moving, twisting a lot more. And wasn't near as conscious of it and was probably doing a lot more with the VR headset then I would have done myself through my own rehab and stuff like that." (FG 3, P4)

Again, this physiotherapist did not experience any pain or discomfort during or after using VR. These perspectives suggest that the immersive quality of VR could encourage individuals with musculoskeletal shoulder pain to move and engage with rehabilitation with less focus on their pain and associated fear of movement.

## Theme 2: Enjoyment in VR improves motivation to exercise. "It's a fun way to kind of exercise"

Physiotherapists felt that VR could offer an engaging platform for rehabilitation and physical activity over and above more traditional physiotherapy engagement. Physiotherapists acknowledged that prescribed exercise can sometimes be repetitive, and patients often lose interest in rehabilitation early on. When asked about their experience of using VR, most physiotherapists reported having fun and suggested that enjoyment in VR could improve concordance with rehabilitation programs.

"For compliance—if it's fun compliance is going to go up. I think everyone had the same response–a big smile when you do the first bit. Everyone smiles so you know, if you smile when you're doing your exercise, you're more likely to want to do it." (FG 1, P1)

Another physiotherapist felt that time flew by when she was using VR and that she was inclined to exercise for longer and with greater intensity while using VR. She reported that she enjoys being physically active and that VR was an enjoyable platform for physical activity.

"I checked the time because I was wrecked. I was flat out on it. That's the truth. Yeah. And I wouldn't be a gamer. I wouldn't be into play stations or games or anything, but I just think I'm an active person and it's just, you know, it's a fun way to kind of exercise". (FG 3, P4)

Physiotherapists were aware of the general health benefits associated with physical activity, not just for musculoskeletal conditions like shoulder pain. They felt that VR could have a potential role in helping to increase physical activity levels in an enjoyable manner. They suggested that VR could be valuable for "even just basically increasing physical activity levels" (FG

1, P4) and "not just then for a localized pathology, it might be you know for improvement of health in other areas". (FG 3, P2)

## Theme 3: Shoulder conditions that may benefit from VR rehabilitation. "I kept coming back to the instability patients"

Physiotherapists explored which clinical shoulder conditions might benefit most from VR interventions. Physiotherapists were quick to suggest using VR with a cohort of patients with shoulder instability. Physiotherapists were able to imagine how they might use VR to facilitate proprioceptive training by providing sensory feedback and introducing task-specific challenges like following a moving target. One physiotherapist felt that this would be a helpful tool for people who have experienced recurrent dislocation and have difficulty with motor control training.

"I kept coming back to the instability patients. I think with those difficult instability patients with recurrent dislocation who have no idea where their shoulder is, it could be a really nice additional sensory input that you could do lots of work down in safe positions and gradually work up into more "dangerous" positions." (FG 1, P2)

Some physiotherapists stated that they would use VR when working with individuals with multidirectional instability who struggle to engage with rehabilitation. One physiotherapist felt that VR would offer an opportunity to progress rehabilitation to include chaotic movement in an engaging way. There was also an agreement among physiotherapists that younger patients might especially enjoy using the VR technology for rehabilitation.

"for that cohort who are the multidirectional kind of instability or that need a bit more chaos in their movement or the ones that aren't, you know, we see a lot of the ones in the hypermobile spectrum and maybe more in that younger population. We see those quite a lot of those teenage mobility with a history of a subluxation type event that's not a true dislocation, and then it's very hard to engage them in their rehab". (FG 2, P1)

Physiotherapists suggested how VR might help in managing individuals with frozen shoulder. One physiotherapist described how individuals with frozen shoulder often "compensate" for loss of mobility at the shoulder by moving other joints such as the elbow and wrist. He argued that VR may help individuals to be aware of how they are moving their upper limb, through feedback from the VR headset and that this could facilitate re-education of movement.

"with a stiff shoulder (frozen shoulder), you can get good feedback on whether its the shoulder that you're actually rotating or elevating rather than your elbow or your wrist as well" (FG2 P2)

Another physiotherapist suggested using VR to promote general exercise for individuals with frozen shoulder would be beneficial. She stated that she would use VR to motivate people and to monitor exercises levels, without focussing so much on shoulder specific exercise.

"Instead of our frozen shoulders coming to our shoulder specific class, we were going to just take a cohort of them and put step counters on them and give them physical activity advice and get them upping their general exercise while they have the frozen shoulder and then comparing to see whether their outcomes improved in terms of initial function. So, you could use VR and not to do any shoulder exercises. Get them doing some cardio or some other program. So yeah, that would be a nice way of using it." (FG 2, P1)

## Theme 4: Safety concerns and adverse effects associated with using VR. "It's very easy to trip over something when you are engaged in the VR simulation."

Despite having reported that VR shows potential as an intervention for managing shoulder conditions, physiotherapists were slow to recommend VR in its current form and felt that

further research and development would be needed before it could be incorporated into clinical practice.

Physiotherapists expressed concerns about the risk of injury, re-injury and flaring up of current symptoms while using VR. Despite the fact that the Oculus Quest headset incorporates an introductory safety tutorial and asks users to setup a safe boundary before using any of the applications, one physiotherapist was sceptical about patients taking the time to observe these safety features. As a result, he was worried that patients may injure themselves while using VR.

"Some people would rush and maybe not be so observant with things. And it's very easy to trip over something when you can't see it and you are engaged in the VR simulation." (FG 2, P2)

Interestingly, the immersive quality of VR is what physiotherapists felt was one of the biggest risk factors to safety. Based on the results of this study, immersion in VR is both a facilitator to rehabilitation and a risk factor to safety.

"It's very easy get engrossed and then you kinda go for the knockout punch or whatever and you go way past where your boundary is." (FG 1, P3)

Some physiotherapists suggested that supervision of the person using VR is an important consideration and may help to prevent accident and injury. Indeed, supervision was proposed as a way of overcoming the safety concerns that physiotherapists have.

"My wife tried it as well and I was like literally, I was kind of shadowing her around, guiding her away from the walls and TV and window because she was going into it and at one stage I had to duck from a punch." (FG 1, P3)

Physiotherapists reported feeling responsible for the health and safety of their patients and were worried about liability in the event of an accident or injury. They were concerned about the possibility of being subject to an insurance claim if an injury was to result from intervention using VR.

"If someone falls over or they, they fall using it. Where would you be? Would you be covered in terms of your insurance and your indemnity?" (FG 3, P2)

Physiotherapists were not only concerned about causing injury but also aggravating current symptoms due to the potential to "overdo" it when immersed in VR. Again, this was associated with the level of immersion experienced while using VR.

"I got so immersed in the game that grading my effort was hard, it's very hard to pull yourself back." (FG1, P2)

One physiotherapist reported that she was moving and exercising her shoulder in such a way that it caused shoulder pain, and as a result she felt reluctant to risk aggravating symptoms in those patients who are already sensitive to movement.

"I felt I was generating forces around my shoulder that were completely out of proportion of what was needed. And so I really felt that it nearly gave me a little bit of shoulder pain, which I don't have. Isn't it awful? I felt that if you had any type of patient who was remotely sensitive, like centrally sensitized or any type of a sensitization, I would be so slow to use it because I feel that you could really, really sensitize them with it." (FG 3, P5)

Motion sickness is a known side-effect of using VR and affects some people more than others. In this study, four physiotherapists described having an experience of motion sickness. In each case, the physiotherapist recovered quickly, chose a different application, and continued to use VR without further difficulty. Physiotherapists described the types of VR applications most likely to cause motion sickness. Games that involve your virtual body moving forwards, while in actual fact you are stationary appear to cause most difficulty.

"I actually do suffer from motion sickness anyway. Certainly, the type of game that you mentioned like the Jurassic Park one, I found that tough. If my body was moving forwards but my feet were staying still like the rollercoaster type thing like that I wouldn't go there." (FG 1, P1)

One physiotherapist recommended that patients should be seated at first when using VR so that they have an opportunity to adapt to the virtual environment. Again, this physiotherapist had difficulty with the roller coaster simulation.

"I haven't used anything like that before, but I found myself pretty disorientated when I put it on straight away. So I think probably a good recommendation would be for people to be seated, putting it on initially. There's the roller coaster demo and I just had to lay down on the floor after that. I just wasn't able at all." (FG 2, P4)

### Theme 5: Practical considerations for use of VR in clinical practice. "It's just figuring out the practicalities."

Physiotherapists considered various practical factors that would inhibit their use of VR in clinical practice. These included hygiene, comfort, cost of technology and the time it takes to set up. Participants suggested that for use in clinical practice, devices must be cleaned regularly, and face guards are recommended to prevent the user's skin from touching the device. Physiotherapists noted that headsets are heavy which means that head and neck muscles can fatigue quickly during use. Participants felt that the cost of VR technology is a significant factor also.

"It's just figuring out the practicalities. There's just so many kind of things you've got to work through. There's the comfort level, the time of it, the cost of it, everything like that." (FG 1, P1)

One physiotherapist who worked in a hospital setting shared her experience of the logistical challenge of loaning equipment to patients and feared that a similar problem would arise when providing VR headsets to patients for use at home. She stated that it can be challenging when the equipment is not returned to the department and experienced stress associated with following up on previously loaned equipment.

"I definitely would have had the experience in the past, when working in the public sector of giving equipment to patients and just the head wreck of trying to get it back. You know this would be things like you know, TENS machine, Neurotec, ankle weights, you know." (FG 3, P3)

A physiotherapist who worked in a private practice setting was more concerned about the cost of the technology and stated that he would not be happy to give the technology to patients to use at home for fear that it would be broken or not returned.

"I couldn't see myself as a business owner having you know five Oculus's and being comfortable throwing them out to people, you know?" (FG 1, P1)

Some physiotherapists felt that the technology was intuitive and easy to use, while others took time to familiarize themselves with the VR environment and controls. Many physiotherapists were concerned about the process of introducing VR technology to patients in clinical practice. The initial set up of the device and explanation of how to use the technology takes time. Physiotherapists were concerned that patients who were less "tech savvy" may be discouraged before they even start.

"You know, if you had maybe someone that found technology quite intimidating and were very reluctant even to try it, it might be difficult." (FG 3, P3)

In terms of overcoming this barrier, physiotherapists felt that adequate training would help them to feel confident in using the technology, instill confidence in patients who are new to VR, and reduce the therapy time dedicated to setting up of the device.

"So the better that we are like, I know I'll pick it up if I know that I'm quick at it. I know the pitfalls of it. I know when they have it on that I'm good at navigating around it." (FG 2, P1)

## Discussion

The aims of this study were firstly, to explore physiotherapists' beliefs and perspectives of immersive VR as a platform for assessment and rehabilitation in patients with musculoskeletal

shoulder pain, secondly, to identify potential barriers and facilitators to using VR in a musculoskeletal setting and thirdly, to gain insight from clinicians that would inform the development of a VR intervention for the rehabilitation of musculoskeletal shoulder pain. A series of three online focus groups that in total included 16 participants were carried out. Five themes were identified, reflecting physiotherapists beliefs that VR provides novel opportunities for shoulder rehabilitation and may offer new avenues for managing movement-related fear and improving concordance with rehabilitation, especially for people with shoulder instability. Barriers related to safety and practical considerations associated with using VR for shoulder rehabilitation were also identified in the final themes.

The results are presented in a way that does not quantify experiences or perspectives of physiotherapists, but instead provides a general statement about the strength of discussion around themes. Use of numbers in qualitative research can in certain instances add precision to statements about the frequency or typicality of ideas and phenomena.[69] However, in this study due to the flow of interviews, not all participants had an equal opportunity to express a particular point of view.[70] In the focus group interview, when a participant raised a point or opinion other participants sometimes clearly stated their agreement or disagreement while some engaged by simply nodding in agreement. The fact that one participant already stated a point sometimes meant that others did not feel the need to repeat it, but this was not necessarily reflective of their position regarding the point being made. In addition, counting number of participants who stated or agreed with a point assumes that the strength of feeling or perception (or whatever people are agreeing to) between each participant is the same if they 'agree'. Some participants can feel very strongly about an issue, some will be carried along by group consensus and some will feel ambiguous, but all responses may get counted in the same pile. The aim of this study was to capture how the issues are much more nuanced as are participants perceptions.[68]

Having spent time using "off-the shelf "VR technology at home, physiotherapists felt that there was a potential role for VR to change movement and fear avoidance behavior in people who have musculoskeletal shoulder pain. Physiotherapists suggested that because VR is such an immersive experience, it might help "distract" people from their pain and cause them to focus their attention on the task within the VR simulation instead. Indeed, two physiotherapists who participated in the study described how their own experience of movement in the context of musculoskeletal pain improved while using VR because they were so engaged in the technology.

These experiences appear to reflect those described by Hoffman et al [45] who suggested that people have a limited amount of conscious attention and that VR may reduce the conscious attention available to focus on pain. These authors (44) also suggest that immersive "high-tech" VR is more effective at demanding attention and reducing pain than non-immersive "low-tech" VR. In this study, they investigated healthy participants' experience of presence in VR and their experience of experimental pain while using two different VR systems. Participants in the "high-tech" group described a greater sense of presence and when subject to painful thermal stimulation, reported a reduction in pain that was superior to the "low-tech" VR group. This finding suggests that immersive VR competes for our attention when a painful stimulation is present, and that VR analgesia may work via distraction from pain. Physiotherapists who participated in the focus group study proposed a similar mechanism after using VR themselves. The theory of distraction as a mechanism for managing pain and kinesiophobia in VR needs to be tested in people with musculoskeletal shoulder pain.

A recent study by Kelly et al [71] explored the experiences of people with chronic low back pain who used a novel immersive VR intervention. In this study, most participants believed that distraction accounted for the mechanism underlying the pain-relief they experienced

while using VR. One participant described their experience of a boxing game and stated that it had enabled them to undertake movement and exercise in a way that they would normally avoid or find intolerable, mirroring the views of the participants in this current study.

Physiotherapists suggested that VR offered an enjoyable way to exercise and that this could motivate people to engage in rehabilitation programs for shoulder pain. In one focus group interview, a physiotherapist pointed out how most participants smiled when they described their experience of using VR, because VR is fun. He stated that if you are having fun, "then compliance is going to go up". For neurological rehabilitation that often involves extensive repetition of basic tasks, VR has been shown to provide an engaging platform for rehabilitation by making exercise more stimulating [39, 40]. Warland et al [40], explored the feasibility and acceptability of using VR for upper limb rehabilitation following stroke and found that the VR system was a source of motivation for participants. They described how the concept of "time flying" was positively correlated with enjoyment in VR and occurs when a participant is immersed in a goal-directed task in VR.

A physiotherapist in the current study described how time passed quickly "I checked the time because I was wrecked" when using VR for physical activity because of the level of enjoyment associated with it. In addition, this physiotherapist felt that she was physically exerting herself to a higher intensity than she would normally do in her training. A study by Warburton et al [72] has shown that interactive video game cycling, using non-immersive VR leads to metabolic requirements that are significantly higher than those achieved during traditional stationary cycling at matched incremental workloads. In Warburton's study, 14 participants were asked to complete 5-minute sessions of workloads of 25%, 50% and 75% of maximum power output on a stationary bike under two conditions: 1) while listening to music 2) while playing interactive video games. When playing video games, participants demonstrated significantly higher heart rate (26%), energy expenditure (61%) and oxygen consumption (34%). In the current focus group study, physiotherapists acknowledged the importance of general physical activity for people with musculoskeletal shoulder pain. It is possible that VR could facilitate motivation to exercise at higher intensities in this cohort.

Physiotherapists were asked "What specific subgroup of individuals with shoulder pain do you think would be most suited to using VR as part of their management?" Interestingly, many physiotherapists felt that people with shoulder instability, who typically present with movement-related fear and have difficulty engaging with rehabilitation would benefit most. Some physiotherapists also considered that VR may be a useful rehabilitation tool for people with frozen shoulder and rotator-cuff related shoulder pain to help improve shoulder function and physical activity levels. Immersive VR has not yet been tested in a cohort of people with musculoskeletal shoulder pain, but a few studies have investigated the effects of non-immersive VR interventions on pain and function for people with shoulder complaints. Lee et al, [52] investigated the effects of a four week period of non-immersive VR training for people with frozen shoulder (n = 16). Significant improvements in range of movement, strength, and function as assessed using the Constant Murley Score were found in all participants. While this study did not include a control group, it shows that VR is well tolerated and has potential to be used successfully in this cohort. In a small RCT (n = 30) including people with subacromial pain, non-immersive VR (Nintendo Wii™) was compared to traditional home exercise over a period of six weeks [51]. Pain intensity and disability improved significantly in both groups. Again, this shows that for people with subacromial pain a non-immersive VR intervention was well tolerated and at least as effective as traditional home exercise. To date, no research has investigated the effect of VR interventions on functional outcomes in people with shoulder instability.

In this study, physiotherapists expressed significant concern around the safety of using immersive VR, especially in a non-supervised manner. Interestingly, immersion was seen by

physiotherapists as a key factor in distracting people from their symptoms. Here immersion was seen as a key risk factor for safety. Physiotherapists were worried that patients might fall and injure themselves because of the immersive nature of VR. They felt that patients may not pay attention to safety features such as virtual boundaries and bump or trip over objects in the real world. None of the physiotherapists themselves sustained injury in this manner but felt that they required that supervision of family members to "check" them and ensure that they remained clear of objects and furniture. Much of the literature on VR interventions for musculoskeletal rehabilitation uses non-immersive technology, which allows the user to see the real-world environment, therefore reducing the risk of accident or injury. A recent review of 13 studies that explored the effectiveness of immersive VR interventions for managing musculoskeletal pain reported the frequency and nature of adverse events [30]. In this review, no adverse events that involved accident or injury were reported, however, many of the interventions were carried out in a supervised manner or in a seated position. None of the included studies involved whole body standing exercise, in an unsupervised setting.

Physiotherapists were also concerned that VR could lead to aggravation of musculoskeletal shoulder pain due to the uncontrolled nature of the exercise involved. Some participants felt that when they were using VR, they became so engrossed in the game that they were generating high levels of force around the shoulder and that this could lead to aggravation of symptoms in someone who is recovering from a shoulder injury. In the review by Brady et al, [30] only one study reported increased pain in one of their participants after using immersive VR [73]. This study included six people with Complex Regional Pain Syndrome (CRPS) who completed 10 sessions of virtual task-specific exercises in an interactive kitchen environment. One participant withdrew from the study due to an increase in nervous system sensitivity. Kelly et al [71] who explored the experiences of people with CLBP who used an immersive VR intervention reported that some participants experienced muscle soreness and back pain that they attributed to an increase in movement and muscle use while using VR. They also reported that these symptoms were managed easily by modifying the position of use (eg sitting instead of standing) or the duration of use. Currently there is little evidence that using VR causes injury or pain, but this must be tested further in people with musculoskeletal shoulder pain.

Motion sickness is a common side-effect of using VR [30, 74] and was reported by physiotherapists who participated in this focus group study. Symptoms reported were short-lived and all participants were able to continue using VR once they avoided applications that gave them a sense that their body was moving forwards in the virtual environment (rollercoaster simulation). Motion sickness occurs because of a conflict between visual and vestibular system cues [74]. Some people experience motion sickness more than others, and various factors have been proposed to explain this [74]. When considering VR for use in clinical practice, it may be possible to identify people who are most at risk of this side-effect and who may not be suitable for VR interventions. Future VR intervention design should consider minimizing the risk of inducing conflict between vestibular and visual systems to optimize comfort and acceptability of the intervention.

Physiotherapists identified several practical factors that would act as barriers to using VR in clinical practice. Some physiotherapists recognized that the cost of VR technology is significant, and they would be concerned about the risk of patients damaging or not returning the equipment. Physiotherapists also acknowledged that it took time to navigate the VR technology and that helping patients to do the same would take time and careful instruction. They also felt that VR may suit patients who were already "tech savvy" and some considered that younger patients may have greater success with using VR. However, physiotherapists felt that with guidance and practice, they would feel confident in delivering a VR intervention to their patients. Contrary to physiotherapist perspectives, research indicates that older adults can use

VR technology successfully at home. A systematic review by Miller et al [75] investigated the effectiveness of VR interventions for enabling physical activity levels in older adults and people with neurological conditions. As a secondary aim, they explored feasibility of using VR in this cohort and found high levels of retention (>70%) and adherence (>64%) across studies.

The results of this study suggest that while VR was seen as an exciting opportunity and potentially opened up new avenues for managing musculoskeletal shoulder pain, it's very 'newness' created some concern and apprehension. In order for VR interventions to be implemented in clinical practice, research is needed to 1) demonstrate its effectiveness as a platform for rehabilitation in a population of individuals with shoulder pain and 2) demonstrate its safety and feasibility in this group. Physiotherapists want to be reassured that immersive VR is well tolerated, has a low risk profile and is relatively easy to use in clinical practice. Physiotherapists also want to be able to identify those patients (specific pathology, stage of rehabilitation, age, technological literacy) that are likely to have the greatest success with VR before they invest time, energy, and money into using the technology.

## Limitations of the study

One of the limitations of the current study was that all of the physiotherapists who participated in this study were based in the Republic of Ireland. Their experiences and perspectives on the potential role of VR in clinical practice may be influenced by both their clinical and cultural backgrounds. It is possible that physiotherapists from different countries and healthcare systems would have different views, therefore results from this study cannot be generalised to the physiotherapy profession outside Ireland. Physiotherapists who participated in this study did not have access to specialised clinical VR software but instead used VR "off the shelf" applications available on Oculus Quest. While this allowed physiotherapists the flexibility to think about how they would design a VR programme for people with shoulder pain, it meant that they were exposed to applications that were sometimes energetic and fast paced. Perhaps this made physiotherapists feel more apprehensive about the potential to cause injury or flare up of symptoms than would have been the case had slower paced applications, designed for managing musculoskeletal pain been used.

## Future research

This study sought to explore only physiotherapist perspectives on the role of immersive VR as a platform for the assessment and rehabilitation of musculoskeletal shoulder pain. Further exploration of patient experiences when using immersive VR at home is needed to create a full picture of the potential of and barriers to using VR in clinical practice for this population and to inform future development of therapeutic VR applications [53]. Results from this study highlight the importance of future research to investigate safety and feasibility of immersive VR interventions for managing shoulder pain. Case series and small clinical trials should explore early clinical efficacy and acceptability before investing in large-scale RCT studies with long-term follow up.

## Conclusion

This is the first study to explore physiotherapist beliefs and perspectives on virtual reality–supported rehabilitation for the management of musculoskeletal shoulder pain. Physiotherapists were enthusiastic about the potential for VR to support rehabilitation in this population. Many considered how VR could facilitate movement and physical activity in those who had difficulty engaging with rehabilitation due to pain and associated fear avoidance behavior. However, while VR was seen as an exciting avenue for rehabilitation, there was significant apprehension

concerning safety and practicality of using the technology in clinical practice. These findings provide valuable insight into clinician acceptability of immersive VR as a platform for rehabilitation and the need for further research to answer the questions posed by physiotherapists in the current study. This research will inform and contribute to human-centered design of VR-supported interventions for managing musculoskeletal shoulder pain.

## Supporting information

**S1 File.**
(DOCX)

**S2 File.**
(PDF)

## Author Contributions

**Conceptualization:** Niamh Brady, Beate Dejaco, Jeremy Lewis, Karen McCreesh, Joseph G. McVeigh.

**Data curation:** Niamh Brady, Beate Dejaco.

**Formal analysis:** Niamh Brady, Beate Dejaco, Jeremy Lewis, Karen McCreesh, Joseph G. McVeigh.

**Investigation:** Niamh Brady, Beate Dejaco, Karen McCreesh, Joseph G. McVeigh.

**Methodology:** Niamh Brady, Beate Dejaco, Jeremy Lewis, Karen McCreesh, Joseph G. McVeigh.

**Project administration:** Niamh Brady.

**Resources:** Niamh Brady.

**Software:** Niamh Brady.

**Supervision:** Jeremy Lewis, Karen McCreesh, Joseph G. McVeigh.

**Validation:** Jeremy Lewis, Karen McCreesh, Joseph G. McVeigh.

**Writing – original draft:** Niamh Brady.

**Writing – review & editing:** Niamh Brady, Beate Dejaco, Jeremy Lewis, Karen McCreesh, Joseph G. McVeigh.

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
