## [Decision Letter · Decision Letter 0]

6 Dec 2022

PONE-D-22-27818“Within five seconds you can see the potential of this.” Virtual Reality in the rehabilitation of musculoskeletal shoulder conditions: a focus group study.PLOS ONE

Dear Dr. Brady,

Thank you for submitting your manuscript to PLOS ONE. After careful consideration, we feel that it has merit but does not fully meet PLOS ONE’s publication criteria as it currently stands. Therefore, we invite you to submit a revised version of the manuscript that addresses the points raised during the review process.

We look forward to receiving your revised manuscript.

Kind regards,

Callam Davidson

Staff Editor

PLOS ONE

Journal Requirements:

https://journals.plos.org/pl.osone/s/file?id=wjVg/PLOSOne_formatting_sample_main_body.pdf and 

2. We note you have included a table to which you do not refer in the text of your manuscript. Please ensure that you refer to Table 1 in your text; if accepted, production will need this reference to link the reader to the Table.

3. Please remove the quote from the study title.

Please amend your current ethics statement to address the following concerns:

a) Did participants provide their written or verbal informed consent to participate in this study?

Please report your qualitative study according to the COREQ checklist provided at https://www.equator-network.org/reporting-guidelines/coreq/ and provide the relevant completed checklist. In the checklist please include sufficient text excerpted from the manuscript to explain how you accomplished all applicable items.  

Throughout, please place citations within square rather than round brackets.

References: For internet sources, please include the date of citation.

Please label and cite supporting information as outlined in our guidance: https://journals.plos.org/plosone/s/supporting-information

Reviewers' comments:

Reviewer's Responses to Questions

**Comments to the Author**

1. Is the manuscript technically sound, and do the data support the conclusions?

Reviewer #1: Yes

Reviewer #2: Yes

2. Has the statistical analysis been performed appropriately and rigorously? 

Reviewer #1: N/A

Reviewer #2: Yes

3. Have the authors made all data underlying the findings in their manuscript fully available?

Reviewer #1: Yes

Reviewer #2: Yes

4. Is the manuscript presented in an intelligible fashion and written in standard English?

Reviewer #1: Yes

Reviewer #2: Yes

5. Review Comments to the Author

Reviewer #1: Line 1-2: The will of the authors to find a “captivating” title is understandable. However to quote the phrase “Within five seconds you can see the potential of this” in the title it seems to be overestimating the content of the article. In fact as the authors reported in lines 392-394 “Despite having reported that VR shows potential as an intervention for managing shoulder conditions, physiotherapists were slow to recommend VR in its current form and felt that further research and development would be needed before it could be incorporated into clinical practice.”

Line 117-120: In the introduction it does not seem relevant to report all the numerical results and p-values of cited studies. I suggest to remove them

Line 122-124: In the introduction it does not seem relevant to report all the numerical results and p-values of cited studies. I suggest to remove them

Line 126-131: In the introduction it does not seem relevant to report all the numerical results and p-values of cited studies. I suggest to remove them

Line 135-142: In the introduction it does not seem relevant to report all the numerical results and p-values of cited studies. I suggest to remove them

Line 139: In the cited study the p-value of the VR group was 0.027. If the authors want to report the rounded result it has to be 0.03 and not 0.02

Line 140: There is not a space between (p=0.4) and (53). There is not the point mark after (53)

Line 140: Reference 53 does not seem to be the correct one relative to the text

Line 145: Authors did not insert the reference number but just the word (ref)

Line 173: There is not a space between “checklist” and “(59)”

Line 203-204: “Participants were asked to use the headset at least three times for a maximum 20-minute duration in each session” Could the author provide a reference for this choice or at least motivate it?

Line 295: “Several” Provide the exact number of physiotherapists who have reported it

Line 303: “A number” Provide the exact number of physiotherapists who have reported it

Line 334: “Most” Provide the exact number of physiotherapists who have reported it

Line 354: “Most” Provide the exact number of physiotherapists who have reported it

Line 363: “Some” Provide the exact number of physiotherapists who have reported it

Line 373: “Less commonly” Provide the exact number of physiotherapists who have reported it

Line 383: There is not the quotation mark at the beginning of the phrase

Line 395: “risk or injury”; are “risk of injury” the correct words?

Line 432: “Some” Provide the exact number of physiotherapists who have reported it

Line 540: There is not the bracket after 61%

Line 555: Reference (30) does not seem to be the correct one relative to the text

Line 556: Does not seem relevant to report the p-values of this cited study. I suggest to remove it

Line 560: Does not seem relevant to report the p-values of this cited study. I suggest to remove it

Line 870: “with question 7”; is it correct or it has to be “with question 6”?

This study does not seem to have a great impact on the topic but it represents the initial step to take into consideration the clinicians’ opinions, which should be an important aspect in physiotherapy, that is instead usually skipped by the researchers.

The article is well written, and although it includes a small number of participants, it could give indications to researchers on how to develop or direct their studies on this topic.

Reviewer #2: The authors have very well enlightened regarding a innovative, sophisticated physical therapy approach that is virtual reality application in musculoskeletal shoulder pain. Traditional physical therapy techniques although have their positive impacts yet owing to the specificity and reliable qualitative approaches there implies a need for more better physical therapy approaches which the authors have evaluated among physiotherapists use. Also with any new approaches there lies a need for continual monitoring of the technique and its efficacies, which the authors have tried to evaluate the barriers with use of the VR. Lastly, in recent times the prevalence of musculoskeletal pain locations have been very common towards shoulder pain on the basis of occupational regular stress and strain which the authors have targeted as their population.

6. PLOS authors have the option to publish the peer review history of their article (what does this mean?). If published, this will include your full peer review and any attached files.

Reviewer #1: **Yes: **Matteo Locatelli

Reviewer #2: **Yes: **Pratik Phansopkar

---

## [Author Response · Author response to Decision Letter 0]

19 Jan 2023

Dear Editor,

Thank you for the opportunity to submit a revised version of the manuscript attached. Many thanks also for your helpful comments.

1. I have updated the style for the title, author, affiliation page according to the link you sent. Unfortunately the first link (https://journals.plos.org/pl.osone/s/file?id=wjVg/PLOSOne_formatting_sample_main_body.pdf) did not work and I was unable to find this information on your website. Please let me know if you wish for me to make further changes to the main body of the manuscript. 

2. I have now referred to the table in question in the body of the text. 

3. I have removed the quote and updated the study title. 

4. I have amended my ethics statement to report how written consent was obtained electronically. I have addressed this in the manuscript and in the ethics statement during the submission process. 

5. I have completed and attached the COREQ checklist including relevant excerpts from the text as well as page numbers where points are addressed. Page numbers are based on the "Manuscript" with track changes. 

6. Citations are now placed in square brackets.

7. For the internet source in my reference list, I have added date of citation. 

I hope that our revisions are satisfactory. Please let me know if you require further amendments or information. 

Yours Sincerely,

Niamh Brady

Reviewer 1

Dear Reviewer, 

Thank you for your helpful insights and recommendations that will only help to strengthen the study. We hope that you will be satisfied with our revised version. Please see below for a detailed response to each point you have made:

Line 1-2: The will of the authors to find a “captivating” title is understandable. However to quote the phrase “Within five seconds you can see the potential of this” in the title it seems to be overestimating the content of the article. In fact as the authors reported in lines 392-394 “Despite having reported that VR shows potential as an intervention for managing shoulder conditions, physiotherapists were slow to recommend VR in its current form and felt that further research and development would be needed before it could be incorporated into clinical practice.” 

Thank you for your comment. We hoped to choose a quote that would reflect the essence of the study but understand what you say about this one potentially overstating the enthusiasm for VR. The quotation in the title has now been removed. 

Line 117-120: In the introduction it does not seem relevant to report all the numerical results and p-values of cited studies. I suggest to remove them 

Thank you. Numerical results have been removed. 

Line 122-124: In the introduction it does not seem relevant to report all the numerical results and p-values of cited studies. I suggest to remove them 

Thank you. Numerical results have been removed.

Line 126-131: In the introduction it does not seem relevant to report all the numerical results and p-values of cited studies. I suggest to remove them 

Thank you. Numerical results have been removed.

Line 135-142: In the introduction it does not seem relevant to report all the numerical results and p-values of cited studies. I suggest to remove them 

Thank you. Numerical results have been removed.

Line 139: In the cited study the p-value of the VR group was 0.027. If the authors want to report the rounded result it has to be 0.03 and not 0.02 

Thank you. Numerical results have been removed.

Line 140: There is not a space between (p=0.4) and (53). There is not the point mark after (53)

Line 140: Reference 53 does not seem to be the correct one relative to the text 

Thank you. Numerical results and the incorrect citation have been removed. 

Line 145: Authors did not insert the reference number but just the word (ref) 

Thank you for pointing this out. There is no reference for this statement, rather an expansion of the point in the following sentence. The word (ref) has been removed.

Line 173: There is not a space between “checklist” and “(59)” 

Thank you. A space has been added here. 

Line 203-204: “Participants were asked to use the headset at least three times for a maximum 20-minute duration in each session” Could the author provide a reference for this choice or at least motivate it? 

Thank you. The Oculus Quest safety instructions advise using VR for a maximum 30-minute duration. We decided to recommend a 20-minute limit as all participants were new to the technology and we felt that this would minimise fatigue and potential motion sickness during a short trial period like this We felt that three 20-minute sessions would allow sufficient time to experience VR and engage with the focus group discussion. We have added justification in the paper’s text now. 

Line 295: “Several” Provide the exact number of physiotherapists who have reported it 

Thank you for your suggestion and we understand how you may value clarity on numbers of physiotherapists reporting perceptions. 

On this matter, we believe that adding numbers of participants here may detract from the inductive process of the qualitative analysis. In a focus group interview, when a participant raises a point or opinion others may state their agreement or disagreement while others may engage by simply nodding in agreement. The fact that one person stated a point sometimes means that others may not feel the need to repeat it, but this is not necessarily reflective of their position regarding the point being made. 

In addition, counting number of participants who stated or agreed with a point assumes that that the strength of feeling or perception (or whatever people are agreeing to) between each participant is the same if they ‘agree’. Some participants can feel very strongly about an issue, some will be carried along by group consensus and some will feel ambiguous, but all responses may get counted in the same pile. Our aim is to capture how the issues are much more nuanced as are participants perceptions. We believe that this is more in line with qualitative research practices. 

Line 303: “A number” Provide the exact number of physiotherapists who have reported it 

Thank you, we have included the specific number of participants here as it is a specific detail, not open to perception and we believe that it adds clarity to the story. 

Line 334: “Most” Provide the exact number of physiotherapists who have reported it 

As above, we believe that adding a specific number here does not add to the presentation of results and does not accurately reflect the strength of the discussion around this point of view. Our preference would be to not use numbers in this instance. 

Line 354: “Most” Provide the exact number of physiotherapists who have reported it 

As above, we feel that the use of numbers is not helpful here but have instead revised the wording of the sentence. 

Line 363: “Some” Provide the exact number of physiotherapists who have reported it 

As above, we feel that the use of numbers is not helpful here. In the discussion this point was intertwined with the discussion around using VR for improving motivation to exercise and therefore, it is difficult to separate the numbers of participants suggesting each opinion/perception regarding the use of VR in clinical practice. 

Line 373: “Less commonly” Provide the exact number of physiotherapists who have reported it 

Again, we prefer not to add specific numbers here but instead have revised the wording.

Line 383: There is not the quotation mark at the beginning of the phrase 

Thank you. This has now been added. 

Line 395: “risk or injury”; are “risk of injury” the correct words? 

Thank you. This has now been corrected. 

Line 432: “Some” Provide the exact number of physiotherapists who have reported it 

Thank you. In this case we agree that including a number is appropriate as this is a specific finding reported by physiotherapists. 

Line 540: There is not the bracket after 61% 

Thank you. I have added a bracket here. 

Line 555: Reference (30) does not seem to be the correct one relative to the text 

Thank you. This was inserted by mistake. It has now been removed. 

Line 556: Does not seem relevant to report the p-values of this cited study. I suggest to remove it 

Thank you. Numerical results have been removed.

Line 560: Does not seem relevant to report the p-values of this cited study. I suggest to remove it 

Thank you. Numerical results have been removed.

Line 870: “with question 7”; is it correct or it has to be “with question 6”? 

Thank you. This has been corrected this to say “with question 6”

Reviewer 2

Dear Reviewer, 

Thank you for your kind words and helpful comments regarding our recent study. We hope that this study will help to further the development and evaluation of virtual reality rehabilitation for musculoskeletal shoulder pain in future years. 

Yours Sincerely, 

Niamh Brady

---

## [Decision Letter · Decision Letter 1]

5 Feb 2023

PONE-D-22-27818R1Physiotherapist beliefs and perspectives on virtual reality supported rehabilitation for the management of musculoskeletal shoulder pain: a focus group study.PLOS ONE

Dear Dr. Brady,

Thank you for submitting your manuscript to PLOS ONE. After careful consideration, we feel that it has merit but does not fully meet PLOS ONE’s publication criteria as it currently stands. Therefore, we invite you to submit a revised version of the manuscript that addresses the points raised during the review process.

We look forward to receiving your revised manuscript.

Kind regards,

Walid Kamal Abdelbasset, Ph.D.

Academic Editor

PLOS ONE

Journal Requirements:

Reviewers' comments:

Reviewer's Responses to Questions

**Comments to the Author**

1. If the authors have adequately addressed your comments raised in a previous round of review and you feel that this manuscript is now acceptable for publication, you may indicate that here to bypass the “Comments to the Author” section, enter your conflict of interest statement in the “Confidential to Editor” section, and submit your "Accept" recommendation.

Reviewer #1: (No Response)

2. Is the manuscript technically sound, and do the data support the conclusions?

Reviewer #1: Yes

3. Has the statistical analysis been performed appropriately and rigorously? 

Reviewer #1: N/A

4. Have the authors made all data underlying the findings in their manuscript fully available?

Reviewer #1: Yes

5. Is the manuscript presented in an intelligible fashion and written in standard English?

Reviewer #1: Yes

6. Review Comments to the Author

Reviewer #1: Line 118: There is not a space between “follow up” and “[46-48]”

Line 131: Still present the “(p<0.05)” I suggest to remove it.

“On this matter, we believe that adding numbers of participants here may detract from the inductive process of the qualitative analysis. In a focus group interview, when a participant raises a point or opinion others may state their agreement or disagreement while others may engage by simply nodding in agreement. The fact that one person stated a point sometimes means that others may not feel the need to repeat it, but this is not necessarily reflective of their position regarding the point being made.

In addition, counting number of participants who stated or agreed with a point assumes that that the strength of feeling or perception (or whatever people are agreeing to) between each participant is the same if they ‘agree’. Some participants can feel very strongly about an issue, some will be carried along by group consensus and some will feel ambiguous, but all responses may get counted in the same pile. Our aim is to capture how the issues are much more nuanced as are participants perceptions. We believe that this is more in line with qualitative research practices.”

I found the arguments of the authors correct and understandable. It would be helpful to somehow put these motivations into the discussion.

7. PLOS authors have the option to publish the peer review history of their article (what does this mean?). If published, this will include your full peer review and any attached files.

Reviewer #1: **Yes: **Matteo Locatelli

---

## [Author Response · Author response to Decision Letter 1]

28 Feb 2023

Dear Reviewer, 

Thank you for your helpful insights and recommendations. We hope that you will be satisfied with our revised version. Please see below for a detailed response to each point you have made:

Line 118: There is not a space between “follow up” and “[46-48]” 

Thank you for your comment. This has now been corrected.

Line 131: Still present the “(p<0.05)” I suggest to remove it. 

Thank you. These numerical results have now been removed. 

“On this matter, we believe that adding numbers of participants here may detract from the inductive process of the qualitative analysis. In a focus group interview, when a participant raises a point or opinion others may state their agreement or disagreement while others may engage by simply nodding in agreement. The fact that one person stated a point sometimes means that others may not feel the need to repeat it, but this is not necessarily reflective of their position regarding the point being made.

In addition, counting number of participants who stated or agreed with a point assumes that that the strength of feeling or perception (or whatever people are agreeing to) between each participant is the same if they ‘agree’. Some participants can feel very strongly about an issue, some will be carried along by group consensus and some will feel ambiguous, but all responses may get counted in the same pile. Our aim is to capture how the issues are much more nuanced as are participants perceptions. We believe that this is more in line with qualitative research practices.”

I found the arguments of the authors correct and understandable. It would be helpful to somehow put these motivations into the discussion. 

Thank you for your comment. We have now added a brief explanation of our approach to the results section and a more detailed discussion of our motivations to the discussion.

Yours Sincerely, 

Niamh Brady

---

## [Decision Letter · Decision Letter 2]

6 Mar 2023

PONE-D-22-27818R2Physiotherapist beliefs and perspectives on virtual reality supported rehabilitation for the management of musculoskeletal shoulder pain: a focus group study.PLOS ONE

Dear Dr. Brady,

Thank you for submitting your manuscript to PLOS ONE. After careful consideration, we feel that it has merit but does not fully meet PLOS ONE’s publication criteria as it currently stands. Therefore, we invite you to submit a revised version of the manuscript that addresses the points raised during the review process.

We look forward to receiving your revised manuscript.

Kind regards,

Walid Kamal Abdelbasset, Ph.D.

Academic Editor

PLOS ONE

Journal Requirements:

Reviewers' comments:

Reviewer's Responses to Questions

**Comments to the Author**

1. If the authors have adequately addressed your comments raised in a previous round of review and you feel that this manuscript is now acceptable for publication, you may indicate that here to bypass the “Comments to the Author” section, enter your conflict of interest statement in the “Confidential to Editor” section, and submit your "Accept" recommendation.

Reviewer #1: (No Response)

2. Is the manuscript technically sound, and do the data support the conclusions?

Reviewer #1: Yes

3. Has the statistical analysis been performed appropriately and rigorously? 

Reviewer #1: N/A

4. Have the authors made all data underlying the findings in their manuscript fully available?

Reviewer #1: Yes

5. Is the manuscript presented in an intelligible fashion and written in standard English?

Reviewer #1: Yes

6. Review Comments to the Author

Reviewer #1: The manuscript provided in the format "Revised Manuscript with Track Changes" is now acceptable for publication.

However, the latest changes made by the authors are not reported in the "Manuscript" format.

7. PLOS authors have the option to publish the peer review history of their article (what does this mean?). If published, this will include your full peer review and any attached files.

Reviewer #1: **Yes: **Matteo Locatelli

---

## [Author Response · Author response to Decision Letter 2]

8 Mar 2023

Dear Reviewer, 

Thank you for reviewing the most recent draft of our manuscript with track changes. I failed to upload the correct version of the “Manuscript” and have now added the most recent version, without track changes. Thanks again for your input and helpful recommendations to date, which have certainly strengthened our paper. 

Yours Sincerely, 

Niamh Brady

---

## [Decision Letter · Decision Letter 3]

3 Apr 2023

Physiotherapist beliefs and perspectives on virtual reality supported rehabilitation for the management of musculoskeletal shoulder pain: a focus group study.

PONE-D-22-27818R3

Dear Dr. Brady,

We’re pleased to inform you that your manuscript has been judged scientifically suitable for publication and will be formally accepted for publication once it meets all outstanding technical requirements.

Kind regards,

Joshua Robert Zadro, PhD

Academic Editor

PLOS ONE

Additional Editor Comments (optional):

As the new handling editor, I have gone back through the past reviewer and editor comments and am happy with how the authors have address all the queries.

Reviewers' comments:

Reviewer's Responses to Questions

**Comments to the Author**

1. If the authors have adequately addressed your comments raised in a previous round of review and you feel that this manuscript is now acceptable for publication, you may indicate that here to bypass the “Comments to the Author” section, enter your conflict of interest statement in the “Confidential to Editor” section, and submit your "Accept" recommendation.

Reviewer #1: All comments have been addressed

2. Is the manuscript technically sound, and do the data support the conclusions?

Reviewer #1: Yes

3. Has the statistical analysis been performed appropriately and rigorously? 

Reviewer #1: N/A

4. Have the authors made all data underlying the findings in their manuscript fully available?

Reviewer #1: Yes

5. Is the manuscript presented in an intelligible fashion and written in standard English?

Reviewer #1: Yes

6. Review Comments to the Author

Reviewer #1: (No Response)

7. PLOS authors have the option to publish the peer review history of their article (what does this mean?). If published, this will include your full peer review and any attached files.

Reviewer #1: **Yes: **Matteo Locatelli

---

## [Editor Report · Acceptance letter]

6 Apr 2023

PONE-D-22-27818R3 

Physiotherapist beliefs and perspectives on virtual reality supported rehabilitation for the management of musculoskeletal shoulder pain: a focus group study. 

Dear Dr. Brady:

I'm pleased to inform you that your manuscript has been deemed suitable for publication in PLOS ONE. Congratulations! Your manuscript is now with our production department. 

Kind regards, 

on behalf of

Dr. Joshua Robert Zadro 

Academic Editor

PLOS ONE